# Beneficial Effect of Fenofibrate in Combination with Silymarin on Parameters of Hereditary Hypertriglyceridemia-Induced Disorders in an Animal Model of Metabolic Syndrome

**DOI:** 10.3390/biomedicines13010212

**Published:** 2025-01-16

**Authors:** Jan Soukop, Ludmila Kazdová, Martina Hüttl, Hana Malínská, Irena Marková, Olena Oliyarnyk, Denisa Miklánková, Soňa Gurská, Zuzana Rácová, Martin Poruba, Rostislav Večeřa

**Affiliations:** 1Department of Pharmacology, Faculty of Medicine and Dentistry, Palacky University Olomouc, 77515 Olomouc, Czech Republic; jan.soukop01@upol.cz (J.S.);; 2Centre for Experimental Medicine, Institute for Clinical and Experimental Medicine, 14021 Prague, Czech Republic; 3Institute of Molecular and Translational Medicine, Faculty of Medicine and Dentistry, Palacky University Olomouc, 77515 Olomouc, Czech Republic

**Keywords:** fenofibrate, silymarin, triglycerides, metabolic syndrome, oxidative stress, insulin resistance, glycogen synthesis, reesterification of fatty acids

## Abstract

**Background:** Hypertriglyceridemia has serious health risks such as cardiovascular disease, type 2 diabetes mellitus, nephropathy, and others. Fenofibrate is an effective hypolipidemic drug, but its benefits for ameliorating disorders associated with hypertriglyceridemia failed to be proven in clinical trials. **Methods:** To search for possible causes of this situation and possibilities of their favorable influence, we tested the effect of FF monotherapy and the combination of fenofibrate with silymarin on metabolic disorders in a unique model of hereditary hypertriglyceridemic rats (HHTg). **Results:** Fenofibrate treatment (100 mg/kg BW/day for four weeks) significantly decreased serum levels of triglyceride, (−77%) and free fatty acids (−29%), the hepatic accumulation of triglycerides, and the expression of genes encoding transcription factors involved in lipid metabolism (Srebf2, Nr1h4. Rxrα, and Slco1a1). In contrast, the hypertriglyceridemia-induced ectopic storage of lipids in muscles, the heart, and kidneys reduced glucose utilization in muscles and was not affected. In addition, fenofibrate reduced the activity of the antioxidant system, including Nrf2 expression (−35%) and increased lipoperoxidation in the liver and, to a lesser extent, in the kidneys and heart. Adding silymarin (micronized form, 600 mg/kg BW/day) to fenofibrate therapy increased the synthesis of glycogen in muscles, (+36%) and reduced hyperinsulinemia (−34%). In the liver, it increased the activity of the antioxidant system, including PON-1 activity and Nrf2 expression, and reduced the formation of lipoperoxides. The beneficial effect of combination therapy on the parameters of oxidative stress and lipoperoxidation was also observed, to a lesser extent, in the heart and kidneys. **Conclusions:** Our results suggest the potential beneficial use of the combination of FF with SLM in the treatment of hypertriglyceridemia-induced metabolic disorders.

## 1. Introduction

Epidemiological studies show that hypertriglyceridemia is increasingly prevalent worldwide, reaching 10% to 25% in the adult population [1,2]. The cause of elevated triglyceride levels may have a genetic background, but most often, it is a combination of polygenic predisposition and unhealthy lifestyle factors, such as nutritional habits, a lack of exercise, and obesity [3,4]. Hypertriglyceridemia also carries serious health risks. Multiple epidemiological studies report associations between cardiovascular disease and circulating triglycerides carried in chylomicrons and very-low density lipoproteins (VLDL), collectively termed triglyceride-rich lipoproteins, (TRL) [5]. There is increasing evidence that TRLs are actually more atherogenic than low-density lipoprotein (LDL), and they increase vascular inflammation and promote atherosclerosis [6,7,8].

Another serious complication of hypertriglyceridemia is the increased ectopic triglyceride deposition in non-adipose tissues. Elevated lipid accumulation is associated with a variety of complications: In the liver, it leads to steatosis and insulin resistance [9]; in the heart, it can induce lipotoxic cardiomyopathy [10]. Increased triglycerides in muscles are associated with muscle insulin resistance, which contributes to the development of metabolic syndrome [11]. In the kidneys, it can cause nephropathy [12]. For this reason, lowering plasma TG is a therapeutic goal to reduce the risk of cardiovascular and other diseases.

Fibrates are one of the most commonly used drugs for hypertriglyceridemia treatment. Their hypolipidemic activity occurs predominantly through peroxisome proliferator-activated receptor alpha (PPARα) activation, which regulates intracellular lipid metabolism through direct transcriptional control of the genes involved in peroxisomal and mitochondrial β-oxidation, fatty acid transport, lipoprotein lipase activation, and triglyceride catabolism [13,14].

Fenofibrate (FF) is a representative of the fibrate drugs that has been the most commonly used in recent years in clinical practice and its effects have also been extensively studied in a number of clinical trials. However, several randomized controlled trials showed that despite a decrease in blood triglyceride levels of approximately 26%, there was no significant decrease in cardiovascular events [15,16,17]. Fenofibrate therapy also did not reduce disorders involved in the development of metabolic syndrome and its complications [18,19]. Moreover, adverse effects of fibrate therapy, such as liver and kidney injury, especially in the presence of high levels of human C-reactive protein (CRP), have been observed [20,21,22]. Adverse side effects associated with renal events and the risk of venous thromboembolism have also been found after treatment with pemafibrate, a new PPARα-selective drug [23]. Research efforts are, therefore, focused on finding additional compounds with the potential to increase the efficacy and mitigate the adverse effects induced by FF therapy.

One such compound, which is a suitable supplement to this type of hypolipidemic therapy, is silymarin (SLM), a standardized polyphenolic extract from the milk thistle (Silybum marianum). According to current knowledge and our own published results, SLM has hepatoprotective, cardioprotective, antioxidant, antidiabetic, and anti-inflammatory effects due to the modulation of various transcription factors, antioxidant enzymes, receptors, and signaling pathways [24,25,26,27].

The hepatoprotective properties of SLM have been utilized in the treatment of conditions such as hepatic steatosis, amatoxin poisoning, and transplantation complications [28,29]. In earlier research, we found that SLM decreased plasma VLDL cholesterol levels, increased the activity of superoxide dismutase (SOD) and glutathione levels (GSH), and reduced the production of lipid peroxides in the liver of rats fed a high-sucrose diet [24]. The therapeutic effects of SLM are limited by its low absorption from the gastrointestinal tract and rapid excretion. However, using micronized extracts increased its bioavailability and in rats with metabolic syndrome, it had more pronounced effects on the increase in HDL cholesterol and improved glucose metabolism compared to the standard extract [25,26].

We hypothesized that the combined therapy of FF with SLM could reduce disorders associated with hypertriglyceridemia such as ectopic lipid accumulation, insulin resistance, oxidative stress, and inflammation in tissues. Previous studies have focused mainly on the effects of FF and SLM monotherapy on lipid metabolism disorders. In the pathogenesis of CVD and T2DM, not only does hypertriglyceridemia play a key role, but so do its lipotoxic mechanisms, such as ectopic triglyceride accumulation, insulin resistance, inflammation, and oxidative stress. We, therefore, hypothesize that the addition of SLM, which has proven antioxidative, anti-inflammatory, and antifibrotic properties, could favorably affect the disorders induced by current or even previously persistent hypertriglyceridemia in HHTg rats. The therapeutic effects of combined therapy of FF and SLM on disorders associated with the lipotoxicity of hypertriglyceridemia have not yet been investigated and there is insufficient evidence of their interaction.

To study the therapeutic effects of FF and its combination with SLM, we used a unique model of hereditary hypertriglyceridemic rats (HHTg). This strain of rats was selected for basal and sucrose-stimulated blood triglyceride levels and mimics the most common cause of hypertriglyceridemia in humans, which is, as mentioned, a combination of genetic and lifestyle factors. HHTg rats exhibit most symptoms associated with metabolic syndrome in humans: hypertriglyceridemia, liver steatosis, muscle lipid accumulation, muscle and adipose tissue insulin resistance, hyperinsulinemia, impaired glucose tolerance, higher serum CRP, and mildly elevated blood pressure. This rat strain is an accepted model of metabolic syndrome [30,31,32], which is not associated with excessive obesity or fat intake.

In this study, we investigated the effect of FF monotherapy and the combined therapy of FF with SLM on metabolic disorders caused by genetically determined hypertriglyceridemia such as ectopic lipid accumulation, impaired glucose utilization in muscle and adipose tissue, oxidative stress in the liver, heart, and kidneys, chronic inflammation, and the expression of genes involved in lipid metabolism. The therapeutic effects of FF with SLM have not yet been tested and there are no data on their mutual interaction.

## 2. Materials and Methods

### 2.1. Animals and Diets

All animal experiments were carried out using 4-month-old male hereditary hypertriglyceridemic (HHTg) rats, provided by the Institute for Clinical and Experimental Medicine (Prague, Czech Republic). This strain of rats was selected from Wistar rats based on a higher sensitivity to sucrose-induced triglyceride formation. HHTg rats are characterized by a higher synthesis of triglycerides in the liver, their increased concentrations in circulation and ectopic accumulation in the liver, muscle, and kidneys. As mentioned in the introduction, other disorders associated with the metabolic syndrome were also demonstrated, such as the resistance of muscle and adipose tissue to the action of insulin, hyperinsulinemia, impaired glucose tolerance, oxidative stress in tissues, and the activation of inflammatory parameters. Detailed characteristics of this rat strain are described [30,32,33,34].

The rats (*n* = 19) were maintained in a 12 h/12 h light–dark cycle at a temperature of 22–25 °C with free access to food and water. They were fed a standard laboratory diet without supplementation (control group, *n* = 6) or supplemented with micronized FF in a dose of 100 mg/kg of body weight per day (FF, *n* = 6) or supplemented with the same dose of FF in a combination with micronized silymarin in a dose of 600 mg/kg body weight per day (FF + SLM, *n* = 7) for 4 weeks. The reason for using the micronized form of SLM is the knowledge of the low bioavailability of standardized extracts caused by low absorption from the gastrointestinal tract and rapid excretion from the body. We previously found that the micronized form of SLM or its component silybin caused the highest increase in HDL cholesterol levels and the greatest decreased glycemia and insulinemia in the animal model of metabolic syndrome compared to standard or phytosome forms [26]. The dose and form of SLM used were chosen based on our previous experimental data [22,24]. The standard laboratory diet contained 23% protein, 43% starch, 7% fat, 5% fiber, and 1% vitamin and mineral mixture (Bonagro, Blazovice, Czech Republic). The doses of FF and SLM were chosen according to the doses reported in the literature and according to our previous studies [22,26]. Micronized FF (Fenofix^®^, Teva Pharmaceuticals, Prague, Czech Republic) was purchased from Ingers Industrial Solutions (Brno, Czech Republic). Micronized SLM was purchased from Favea (Koprivnice, Czech Republic). Daily food intake was measured throughout the duration of the experiment. At the end of the study, the rats were decapitated in a postprandial state and their blood serum and tissues were collected for ex vivo incubation, final biochemical analyzes, or gene expression determination.

All experiments were performed in agreement with the Animal Protection Law of the Czech Republic (501/2020), which complies with the European Community Council recommendations for the use of laboratory animals (86/609/ECC), and approved by the Ethics Committee of the Institute for Clinical and Experimental Medicine, Prague.

### 2.2. Biochemical Analysis

The serum concentration of triglycerides, glucose, and total cholesterol was measured using commercially available kits (Erba-Lachema, Brno, Czech Republic). Non-esterified fatty acid (NEFA) concentrations were determined using an acyl-CoA oxidase-based colorimetric kit (F. Hoffmann-La Roche AG, Basel, Switzerland). Glycerol was measured by kit from Merck (Darmstadt, Germany). Serum insulin concentration was determined using a Rat Insulin ELISA kit from Mercodia (Uppsala, Sweden), HMW adiponectin using ELISA kit from MyBioSource (San Diego, CA, USA), and C-reactive protein concentration using a Rat CRP ELISA kit from Alpha Diagnostics (San Antonio, TX, USA). Serum IL-6 was determined using rat ELISA kits (BioSource International, Inc., Camarillo, CA, USA), and rat MCP-1 by ELISA kits (Instant ELISA eBioscience, Thermo Fisher Scientific, Waltham, MA, USA). AST and ALT enzyme activities were measured spectrophotometrically by routine methods using a kit from Roche.

### 2.3. Oxidative Stress Parameters

The activities of superoxide dismutase (SOD), glutathione peroxidase (GSH-Px), glutathione reductase (GR), and glutathione transferase (GST) were analyzed using Assay kits (Cayman Chemical Company, Ann Arbor, MI, USA). Catalase activity determination was based on the ability of H_2_O_2_ to form a color complex with ammonium molybdate ((NH_4_)_6_Mo_7_O_24_) detected spectrophotometrically. The concentration of conjugated dienes was determined by extraction with a mixture of heptane and the level was analyzed by the reaction with thiobarbituric acid according to Malinska et al. [34]. Paraoxonase-1 (PON-1) activity was determined according to the rate of hydrolysis of paraoxon to p-nitrophenol [35].

### 2.4. Analysis of Triglyceride Concentrations in Tissues

A determination of the concentration of triglycerides in the liver and skeletal muscle was performed using extraction in a mixture of chloroform and methanol. Subsequently, a solution of 2% potassium dihydrogen phosphate (KH_2_PO_4_) was added. After extraction, the organic phase was taken and evaporated under nitrogen gas. The resulting pellet was dissolved in isopropyl alcohol and then, the concentration of triglycerides and cholesterol was measured by an enzymatic assay (Erba-Lachema, Brno, Czech Republic).

### 2.5. Lipolysis in Abdominal Adipose Tissue

The measurement of lipolysis in adipose tissue was carried out as described before [36]. Briefly, distal parts of epididymal adipose tissue were incubated in Krebs-Ringer phosphate buffer with 3% of bovine serum albumin (fraction V) at 37 °C for 2 h and the non-esterified fatty acids (NEFA) and glycerol concentrations in the medium were measured using the kit described.

### 2.6. Glucose Utilization in Muscle

Basal and insulin-stimulated glycogen synthesis and glucose oxidation were determined ex vivo in isolated muscle by measuring the incorporation of ^14^C-U glucose into glycogen and CO_2_ as described previously [37]. Briefly, diaphragmatic muscles were incubated for 2 h in 95% O_2_ + 5% CO_2_ in Krebs-Ringer bicarbonate buffer, pH 7.4, containing 0.1 μCi/mL of ^14^C-U glucose, 5 mmol/L of unlabelled glucose, and 2.5 mg/mL of bovine serum albumin (Merck, Prague, Czech Republic), with or without 250 μU/mL insulin. After the incubation, glycogen was extracted from the tissue and counted for radioactivity. To determine glucose oxidation, 0.3 mL of hyamine hydroxide (Perkin Elmer, Waltham, MA, USA) was injected into the central compartment of the incubation vial and 0.3 mL of 5 M H_2_SO_4_ was added to liberate CO_2_. After the incubation for 45 min, when the trapping took place, the hyamine hydroxide was quantitatively transferred to the vial containing scintillation fluid for measuring of ^14^CO_2_ radioactivity.

### 2.7. Glucose Utilization in Adipose Tissue

Glucose utilization in adipose tissue was determined ex vivo by measuring the incorporation of ^14^C-U-glucose into adipose tissue lipids, as described previously [38]. Briefly, distal parts of epididymal adipose tissue were incubated for 2 h in the same buffer as described above for muscles, in the presence (250 IU/mL) or absence of insulin in incubation media. After the incubation, the estimation of ^14^C-glucose incorporation into neutral lipids was carried out. The pieces of tissues were homogenized, and lipids were extracted (chloroform–methanol 2:1) at 4 °C for 16 h, then, KH_2_PO_4_ was added, an aliquot was evaporated and reconstituted in scintillation liquid, and the radioactivity was measured by scintillation counting.

### 2.8. Gene Expression Assay

For the total mRNA isolation, we used the RNeasy Mini Kit (Qiagen, Valencia, CA, USA). We used the amount of 1 µg of mRNA for cDNA synthesis using a Transcriptor High Fidelity cDNA synthesis kit (F. Hoffmann-La Roche AG, Basel, Switzerland). For the determination, we used TaqMan Gene Expression Assays probes (Thermo Fisher Scientific, Waltham, MA, USA) containing target primers and a sequence-specific probe optimized for the best functional performance. Commercially available primers purchased from the same corporation were used to determine the mRNA of, Srebf1, Srebf2, Nr1h4, Nr1h3, Rxr, Pparα, and Slco1a1. The real-time PCR was performed on 1536-well plates using acoustic liquid handler Echo 550 (Labcyte, Dublin, Ireland) and LightCycler 1536 Instrument (F. Hoffmann-La Roche AG, Basel, Switzerland). We performed the measurements in six parallels. Reverse transcription and quantitative real-time PCR analyses for expression of the Nrf2 gene in the liver were carried out using the TaqMan RNA-to-CTTM 1-Step Kit and the TaqMan Gene Expression Assay (Applied Biosystems, Foster City, CA, USA) and analyzed via the ViiA^TM^ 7 Real-Time PCR System (Applied Biosystems, Foster City, CA, USA). The results were evaluated by the ^ΔΔ^Ct method, and all results were normalized and related to the Hprt1 gene (Thermo Fisher Scientific, Waltham, MA, USA).

### 2.9. Statistical Analysis

The data are expressed as means ± standard error of the mean (SEM). All the data were analyzed using Statistica software (ver. 12.0, 2013, TIBCO Software, Palo Alto, CA, USA). The normality of the data was tested using the Shapiro–Wilk test. We used one-way analyses of variance (ANOVA) at the significance level of *p* < 0.05 followed by the post hoc Bonferroni to avoid false-positive results.

## 3. Results

### 3.1. Effect of Fenofibrate Alone and Fenofibrate in Combination with Silymarin on Body Weight and Metabolic Parameters

As shown in Figure 1, treatment with FF alone reduced the body weight of the rats by 11% (*p* < 0.01) compared to the control group (Figure 1A). The amount of abdominal adipose tissue assessed by epididymal (Figure 1B) and perirenal (Figure 1C) fat pad weight was reduced after FF treatment by 18% and 31%, respectively (both *p* < 0.01), while food intake was not affected by FF. FF treatment reduced serum concentrations of triglycerides by 77% (*p* < 0.001; Figure 1D) and non-esterified fatty acids (NEFA) by 29% (*p* < 0.05; Figure 1E) compared to the untreated control. On the other hand, serum concentrations of glucose (Figure 1F), insulin (Figure 1G), alanine (Figure 1H), and aspartate (Figure 1I) aminotransferases, as well as adiponectin (Figure 1J), were not affected by FF monotherapy.

FF combined with SLM reduced body weight by 7% (*p* < 0.01; Figure 1A) and unexpectedly increased food intake by 12%, which led to a milder reduction of epididymal (−12% Figure 1B) and perirenal (−21% Figure 1C) fat pad weight (both *p* < 0.05) compared to untreated groups. Combined FF therapy with SLM affected serum concentrations of triglycerides, NEFA, and glucose (Figure 1D–F) as much as FF monotherapy.

The addition of SLM to FF therapy resulted in a significant reduction in serum insulin concentration of 34% (*p* < 0.05; Figure 1G) compared with FF therapy alone.

### 3.2. Effect of Fenofibrate Alone and in Combination with Silymarin on Lipid Metabolism

Furthermore, we monitored whether FF alone and in combination with SLM would influence the hypertriglyceridemia-associated ectopic accumulation of lipids in tissues (Figure 2). A moderate decrease in soleus muscle triglycerides of 23% in the group treated with FF did not reach statistical significance. Hepatic triglyceride levels were significantly reduced in FF- (−67%; *p* < 0.01) and FF + SLM (−65%; *p* < 0.01)-treated rats in comparison to the controls. The concentration of triglycerides in the kidneys was not affected by FF or FF + SLM treatment.

We further analyzed whether changes in the concentration of lipids in the blood and tissues were associated with fatty acids released from adipose tissue (Figure 3). We found that FF increased the release of NEFA (+28%; *p* < 0.05) and glycerol (+62%; *p* < 0.01) from abdominal adipose tissue. It is known that a certain percentage of fatty acids after release from adipose tissue is re-esterified and returned to the tissue. We found that the re-esterification of fatty acids into triglycerides, as measured by the NEFA/glycerol ratio, was reduced in FF-treated animals (−22%; *p* < 0.05) and in FF + SLM-treated rats (−17%; NS). Reduced re-esterification of released fatty acids from adipose tissue may be one of the mechanisms involved in the reduced amount of visceral adipose tissue in FF-treated rats.

### 3.3. Effects of Fenofibrate Alone and in Combination with Silymarin on Glucose Metabolism and Insulin Sensitivity

Although elevated postprandial glycemia levels in HHTg rats were not affected by FF treatment alone or the combination of FF with SLM, insulin concentrations were significantly reduced by this therapy (Figure 1F, G). This suggests that glucose utilization in muscle tissue for glycogen synthesis, which represents the majority of postprandial glucose disposal [39], may be affected. Therefore, we measured the effect of FF and FF + SLM treatment on ^14^C-U-glucose incorporation for glycogen synthesis and for oxidation in skeletal muscle.

As shown in Figure 4A, FF treatment significantly decreased basal incorporation of glucose into glycogen (−39%; *p* < 0.05). This adverse effect was partially alleviated by the combination of FF + SLM (+36%, FF + SLM vs. FF; *p* < 0.05). An interesting finding was the increased glucose oxidation in muscle tissue observed in FF- (+50%; *p* < 0.05) and FF + SLM (+40%; *p* < 0.05)-treated groups compared to untreated controls (Figure 4B).

To investigate further the effects of FF on glucose utilization in tissues, we analyzed the basal and insulin-stimulated glucose incorporation into lipids in epididymal adipose tissue (Figure 5). Contrary to our expectations, treatment with FF alone and FF in combination with SLM resulted in a similar increase in basal glucose incorporation into lipids of 30% (both *p* < 0.05) and insulin-stimulated lipid synthesis of 22% and 17% (both NS) compared to the untreated control group.

### 3.4. Effect of Fenofibrate Monotherapy and Combination with Silymarin on Oxidative Stress and Inflammation—Related Parameters

Oxidative stress is a key mediator of cellular damage and inflammation and may contribute to the formation of lipotoxic peroxides, which are involved in the development of the disorders associated with hypertriglyceridemia. We assessed the effect of FF alone and in combination with SLM on hypertriglyceridemia-related oxidative stress in tissues by measuring the activities of antioxidative enzymes, glutathione, and lipoperoxidation products, conjugated dienes, and TBARS. As shown in Table 1, in the liver, FF treatment reduced the expression of Nrf2, an important transcription factor that controls the expression of several genes of antioxidant enzymes. FF therapy also decreased the activity of PON-1 and SOD, and despite the slightly increased activity of GST and catalase, led to increased formation of lipoperoxidation products. Interestingly, the addition of SLM to FF therapy reduced the negative effect of FF as suggested by increasing the activity of SOD, CAT, glutathione-dependent enzymes, and PON-1. Our results further showed that glutathione may play a role in the mechanism of the beneficial effect of SLM on oxidative stress. FF in combination with SLM increased the concentration of the oxidized form of glutathione—GSSG—and the GSH/GSSG ratio, suggesting that it could positively affect the activity of glutathione-dependent enzymes. Further evidence for the positive effect of SLM on oxidative stress was partial to the elimination of FF-induced lipoperoxidation as shown by a reduced TBARS concentration in HHTg rats treated with FF + SLM compared to rats treated with FF alone.

In the heart, FF treatment only increased glutathione reductase activity and the production of the initial products of lipid peroxidation, conjugated dienes. Combined treatment with FF + SLM increased the activity of glutathione dependent enzymes—GSH-Px, GR, and GST—and reduced the formation of the final products of lipoperoxidation by 21% (*p* < 0.05) compared to FF-treated rats (Table 2).

In the kidney cortex, FF therapy led to a slight increase in the activity of antioxidant enzymes GR and GST and to an increase in the production of lipoperoxidation products—conjugated dienes (+70%, *p* < 0.05) and TBARS (+27%, *p* < 0.05)—compared with untreated control. SLM in combination with FF increased GSH-Px activity and reduced the FF-induced increase in conjugated dienes concentration (Table 3).

Oxidative stress appears to be an important mechanism underlying the progression of chronic inflammation and, therefore, we tested the effect of FF and its combination with SLM on serum levels of inflammatory markers. As shown in Table 4, serum concentrations of MCP-1, a chemokine with a pivotal role in the regulation of inflammatory processes by migration and infiltration of monocytes/macrophages, were significantly reduced by FF treatment, while FF in combination with SLM increased concentrations. On the other hand, the FF and FF + SLM treatment had no significant effect on the serum concentrations of inflammatory biomarkers, IL-6 and CRP.

### 3.5. Effects of Fenofibrate Monotherapy and Combination with Silymarin on Gene Expression

To search for the mechanisms responsible for the effects of FF in combination with SLM on lipid metabolism, the expression of genes in the liver was analyzed. Sterol regulatory element binding proteins (SREBPs) are transcription factors that regulate genes involved in cholesterol, fatty acid, and triglyceride synthesis. The gene expression of Srebf1 was not affected by experimental therapy (Figure 6A) but Srebf2 gene expression was decreased in rats treated with FF (−42%; *p* < 0.01) or FF + SLM (−50%; *p* < 0.01) compared to control animals (Figure 6B). The question of whether the effects of SLM are mediated by the activation of PPARα has not yet been clarified. The results shown in Figure 6C show that while PPARα expression was stimulated by FF (+60%; *p* < 0.05), SLM did not affect expression. This is consistent with the fact that FF improves dyslipidemia by mediating PPARα.

There was no difference between groups in the gene expression of Nr1h3 encoding liver X receptor alpha (LXRα) (Figure 6D). In contrast, FF alone and in combination with SLM suppressed Nr1h4 gene expression (−59%; *p* < 0.01), encoding farnesoid X receptor (Figure 6E). The gene expression of retinoid X receptor α (Rxrα) was slightly decreased in rats treated with FF alone and in combination with SLM compared with untreated controls (Figure 6F). Further results showed that FF alone (−42%; *p* < 0.01) and in combination with SLM (−57%; *p* < 0.01) reduced the expression of the Slco1a1, a bile acid uptake transporter gene (Figure 6G).

Next, we analyzed the relative expression of transcription factor Nrf2, which is a regulator of multiple genes involved in protection against oxidative stress. We found (Figure 6H) that FF significantly decreased expression in comparison with untreated controls (−35%; *p* < 0.05), and the addition of SLM to FF therapy partially decreased this negative impact (+18%; *p* = 0.28) compared with FF alone treatment. It can be summarized that treatment with SLM alone did not affect the expression of the analyzed genes compared to the untreated control group. In contrast, FF therapy increased the expression of PPARα and downregulated the gene expression of Srebf2, Nr1h4, Rxrα, Slco1a1, and Nrf2. Combining FF therapy with SLM, despite a slight increase in Nrf2 expression, did not affect the changes in the expression of genes involved in lipid metabolism induced by FF therapy alone.

## 4. Discussion

One of the most commonly used drugs for the treatment of hypertriglyceridemia is fenofibrate. This effectively decreases blood triglycerides, including atherogenic chylomicron and very low-density lipoprotein remnants [40]. In contrast, in well-controlled clinical trials, neither fenofibrate nor pemafibrate reduced cardiovascular disease [15,16,17]. The beneficial effect of lowering triglycerides on the risk of developing metabolic syndrome has also not been clearly proven [41,42]. Moreover, in some studies, fibrate treatment is associated with an increased risk of liver damage and renal dysfunction [20,21,22]. The underlying causes and preventive measures for their elimination remain unclear.

In this study, we investigated the effect of FF monotherapy on metabolic disorders caused by genetically determined hypertriglyceridemia. We further tested whether it is possible to alleviate some adverse effects of FF by combining the therapy of FF with that of SLM, which exhibits antioxidant, anti-inflammatory, and cardioprotective properties [27].

The study was conducted on a unique strain of rats with genetically determined hypertriglyceridemia exhibiting skeletal muscle and liver steatosis, insulin resistance, mild hypertension, and oxidative stress in tissues. This model is currently well established for the study of the pathogenesis and treatment of hypertriglyceridemia and its consequences [30,32,34,38]. In addition, metabolic disorders in HHTg rats resemble a situation comparable to human conditions, where genetic predisposition is the main cause of the hypertriglyceridemia.

Our results showed that in HHTg rats, FF treatment reduced the levels of serum triglycerides, NEFA, and triglyceride accumulation in the liver, which is consistent with the known effects of FF. FF treatment reduced visceral fat depots in HHTg rats as assessed by epididymal and perirenal fat pad weights. The number of triglycerides in adipose tissue is determined by the intake of fatty acids from circulating lipoproteins, de novo synthesis of fatty acids in adipose tissue, and/or release of fatty acids from stored triglycerides in adipocytes. Detailed analysis showed that that FF treatment reduced lipid accumulation in adipose tissue not only by the increased release of fatty acids from adipose tissue but also by the newly observed reduced re-esterification of fatty acids to triglycerides, which is a necessary step before being stored in adipocytes (Figure 3).

The data from this study provide innovative information about the effect of FF on the mechanisms that may contribute to the fact that the hypotriglyceridemic effect of FF does not reduce the risk of CVD and insulin resistance. These include the findings that FF therapy decreased fatty acids re-esterification and their reduced storage as triglycerides in adipose tissue, which may lead to their increased ectopic storage (outside adipose tissue) even in the presence of lower circulating triglyceride levels. This disorder may also result in reduced glucose utilization for muscle glycogen synthesis, which represents the majority of postprandial glucose disposal and contributes to insulin resistance.

Increased accumulation of triglycerides and their intermediates, diacylglycerols and ceramides, in non-adipose tissues is one of the most serious complications associated with hypertriglyceridemia. Increased lipid accumulation in muscle is connected to muscle insulin resistance [43]; in the liver, it leads to insulin resistance [39]; in the kidney, it can cause nephropathy [44]; and in the heart and blood vessels, it can result in reduced ventricular compliance and an increased risk of atherosclerosis [45,46]. Despite these facts, intramuscular lipids are rarely analyzed in studies. Our results showing the inability of FF to reduce ectopic triglyceride storage in muscle in hereditary hypertriglyceridemia are inconsistent with the results in high-fat-fed animal models, in which clofibrate, gemfibrozil, and WY14643 reduced muscle triglycerides [47]. Reduced intramuscular lipid accumulation has also been observed in high-fat-diet-induced obesity in mice treated with FF [48]. In contrast, in clinical studies, FF administration did not affect muscle triglyceride accumulation [18,19], similar to our study. Further studies should verify whether the observed differences between diet and genetically induced obesity can explain the differences in resistance to lipid-lowering therapy.

Our study showed that another mechanism that may influence the inability of FF to reduce glucose metabolism disorders in hypertriglyceridemia is the observed reduced glucose incorporation into muscle glycogen in FF treated rats. The significance of this finding is of primary importance because reduced glycogen synthesis in muscles is the underlying cause of insulin resistance. Reports to date investigating the effects of fenofibrate on insulin resistance are equivocal. While improved insulin sensitivity with fibrate treatment has been generally observed in rodent models [47], it has been virtually never observed in human studies [18,19]. Our findings in HHTg rats are similar to those in humans, where FF did not affect muscle triglyceride concentration [18].

Our results suggest that the reason for the reduced utilization of glucose for glycogen synthesis may be the increased utilization for the oxidation of glucose in muscle as an energy substrate at lower levels of circulating fatty acids.

Oxidative stress and inflammation are considered key mechanisms in the progression of hypertriglyceridemia to insulin resistance, diabetes, and cardiovascular complications [48], but the findings on the effect of FF on oxidative stress in metabolic syndrome and diabetes remain unclear. It has been found that FF treatment can reduce, not affect or even aggravate, oxidative stress [22,49,50]. In this study, we show new findings about the effect of FF therapy on increased oxidative stress in tissues in genetically determined hypertriglyceridemia. HHTg rats treated with FF alone showed increased oxidative stress in the liver, kidney cortex, and, to a small extent, also in the heart. In the liver, FF therapy reduced the activity of SOD and PON-1, and elevated concentrations of lipid peroxidation products. The effect of FF on the decrease in PON-1 activity is an important new finding that could contribute to explaining the inability of FF to reduce cardiovascular disorders. This enzyme hydrolyses oxidized lipids and protects microvascular complications caused by oxidative stress, and its decreased activity has been associated with an increase in the incidence of cardiovascular disease [51,52]. These changes in the antioxidant system were associated with a markedly reduced expression of transcription factor Nrf2, a crucial regulator of the cellular defense against oxidative stress.

Oxidative stress is associated with the induction of inflammation processes. We found that in HHTg rats, FF therapy only reduced serum MCP-1 concentrations, while CRP and IL-6 levels were not affected.

To search for the mechanisms responsible for the lipid-lowering effects of FF, we analyzed the hepatic mRNA expression of genes encoding transcription factors involved in lipid metabolism. FF treatment markedly increased the gene expression of PPARα, which is consistent with the known fact that FF exerts lipid-lowering effects through the upregulation of the transcription factor PPARα that regulates genes involved in fatty acid uptake, as well as their metabolism, and increased lipid oxidation [53]. In contrast, FF downregulated the expression of Nr1h4 encoding farnesoid X receptor, Rxrα, and Slco1a1, a bile acid uptake transporter gene.

Oxidative stress is associated with the induction of inflammation processes. We found that in HHTg rats, FF therapy only reduced serum MCP-1 concentrations, while CRP and IL-6 levels were not affected.

In this study, we provide new findings showing that the addition of SLM to FF therapy alleviates the adverse effects of FF on some metabolic parameters that could reduce the effectiveness of FF in influencing the risks of cardiovascular diseases and insulin resistance. A significant finding was that SLM alleviated the FF-induced adverse effect on muscle glucose utilization for glycogen synthesis (Figure 2). Given that skeletal muscle is the main tissue responsible for tissue glucose utilization and insulin action, and is, therefore, considered a major site for the development of insulin resistance and type 2 diabetes, this finding is important. The beneficial effect of SLM on insulin resistance has also been observed in a clinical trial [54]. A consequence of the positive effect of SLM on glycogen synthesis may be a reduced postprandial insulin level in rats treated with FF in combination with SLM (Table 1).

Another beneficial effect of SLM was the reduction in oxidative stress, especially in the liver, but also in the myocardium and kidneys, as shown by the reduced activity of the antioxidant system and the increased formation of lipid peroxidation products, conjugated dienes, and TBARS. The beneficial effect of SLM on hepatic PON-1 activity is an important new finding because this enzyme hydrolyses oxidized lipids and protects microvascular complications caused by oxidative stress, and its decreased activity has been associated with an increase in the incidence of cardiovascular disease [51,52]. The beneficial effect of SLM on oxidative stress may be related to the antioxidant activity, which may be a consequence of direct free radical scavenging, a reduced production of reactive oxygen substances in mitochondria, and an increase in antioxidant defense via nuclear transcription factor Nrf2, a crucial regulator of the cellular defense against oxidative stress [55,56,57]. Our results show that a marked decrease in Nrf2 expression in FF-treated rats can be attenuated by SLM therapy, although the increase was not statistically significant.

Our study is limited by the small number of animals in the groups (*n* = 6–7). Since HHTg rats are bred in a brother x sister system, variances between animals are minimized.

## 5. Conclusions

Our results show that in a model of genetically determined hypertriglyceridemia, FF monotherapy effectively reduces serum concentrations of triglycerides, NEFA, in the liver and reduced the accumulation of triglycerides and the expression of genes encoding transcription factors involved in lipid metabolism. The results provide new findings that the FF-induced reduction of visceral adipose tissue is a consequence of not only reduced lipolysis but also of the reduced re-esterification of fatty acids and their deposition in adipose tissue. New findings that could partially explain the failure of FF therapy to reduce insulin resistance and cardiovascular disorders are reduced glucose utilization in muscle tissue for glycogen synthesis, and lowered activity of the antioxidant system associated with increased lipid peroxidation in the liver and, to a lesser extent, in heart and kidneys, where ectopic triglyceride accumulation was not reduced by FF. Adding SLM to FF therapy increased the utilization of glucose in muscles, and reduced hyperinsulinemia, which indicates improvement in insulin resistance. In line with the known hepatoprotective effects of SLM, its beneficial effect on lipid metabolism and the antioxidant system was most evident in the liver and less so in the heart and kidneys. Our findings should be the basis for verification in clinical studies. Also, the search for other options to increase the effectiveness of FF therapy by combining it with substances with antioxidant and anti-inflammatory effects to increase the effectiveness of FF therapy in reducing the risk of cardiovascular disorders and type 2 diabetes should be the subject of further studies. Given the suggested differences in the effectiveness of hypolipidemic therapy between dietary and genetically induced hypertriglyceridemia, these differences should be analyzed in more detail.

## Figures and Tables

**Figure 1 biomedicines-13-00212-f001:**
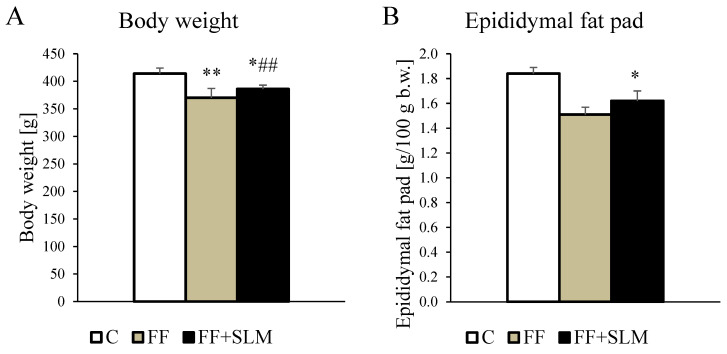
Effects of FF alone and FF in combination with SLM on body weight (**A**) and metabolic parameters (**B**–**J**) in HHTg rats. Data are expressed as mean ± SEM; *n* = 6–7 animals per group. Abbreviations: C—control; FF—fenofibrate; FF + SLM—fenofibrate + silymarin; NEFA—non-esterified fatty acids. Statistically significant differences: * *p* < 0.05; ** *p* < 0.01; *** *p* < 0.001 denote significance versus the control; # *p* < 0.05; ## *p* < 0.01 denote significance versus the FF group.

**Figure 2 biomedicines-13-00212-f002:**
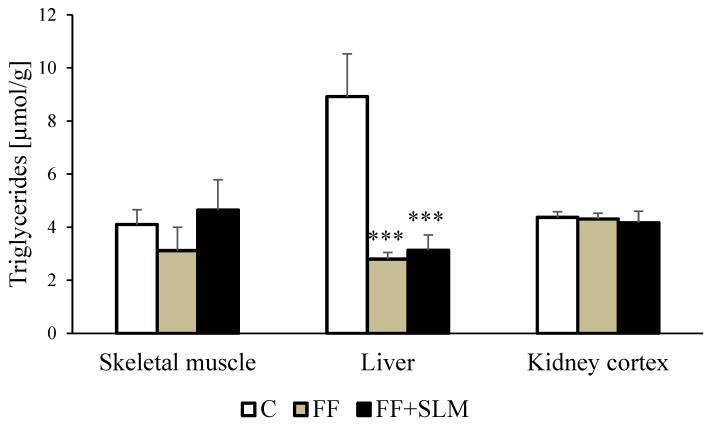
Ectopic accumulation of triglycerides in tissues of HHTg rats treated with FF alone and FF with combination with SLM. Data are expressed as means ± SEM; *n* = 6–7 animals per group. Abbreviations: C—control; FF—fenofibrate; SLM—silymarin. Statistically significant differences: *** denotes *p* < 0.001 significance versus control.

**Figure 3 biomedicines-13-00212-f003:**
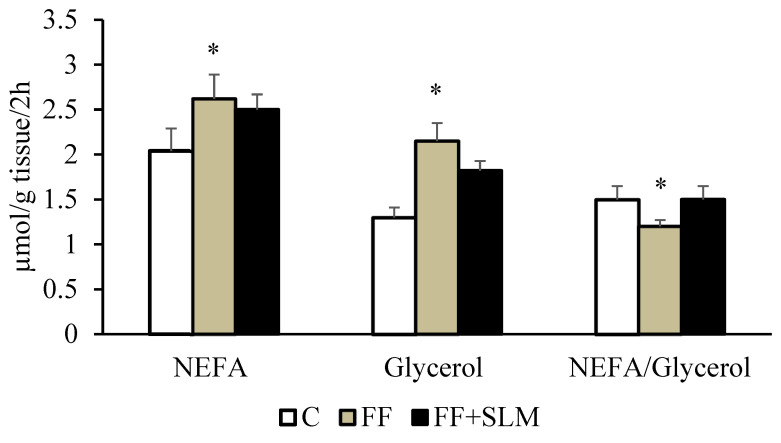
Effect of FF alone and in combination with SLM on adrenaline-stimulated lipolysis in epididymal adipose tissue of HHTg rats. Data are expressed as means ± SEM; *n* = 6–7 animals per group. Abbreviations: C—control; FF—fenofibrate; FF + SLM—fenofibrate + silymarin; NEFA—non-esterified fatty acids. Statistically significant differences: * *p* < 0.05; denotes significance versus control.

**Figure 4 biomedicines-13-00212-f004:**
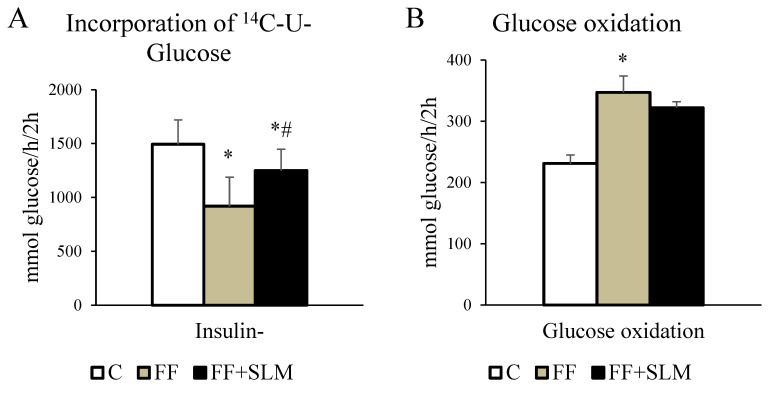
Effect of FF alone and in combination with SLM on basal and insulin-stimulated incorporation of ^14^C-U-Glucose into glycogen (**A**) and into CO_2_ (**B**) in muscle of HHTg rats. Data are expressed as means ± SEM; *n* = 6–7 animals per group. Abbreviations: C—control; FF—fenofibrate; FF + SLM—fenofibrate + silymarin. Statistically significant differences: * *p* < 0.05 denotes significance versus the control; # *p* < 0.05 denotes significance versus the FF group.

**Figure 5 biomedicines-13-00212-f005:**
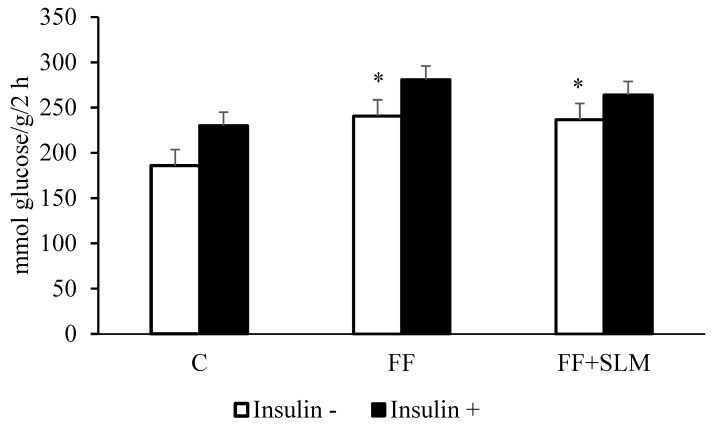
Effect of FF alone and in combination with SLM on basal and insulin-stimulated incorporation of ^14^C-U-Glucose into in epididymal adipose tissue of HHTg rats. Data are expressed as mean ± SEM; *n* = 6–7 animals per group. Abbreviations: C—control; FF—fenofibrate; FF + SLM—fenofibrate + silymarin. Statistically significant differences: * *p* < 0.05; denotes significance versus untreated control.

**Figure 6 biomedicines-13-00212-f006:**
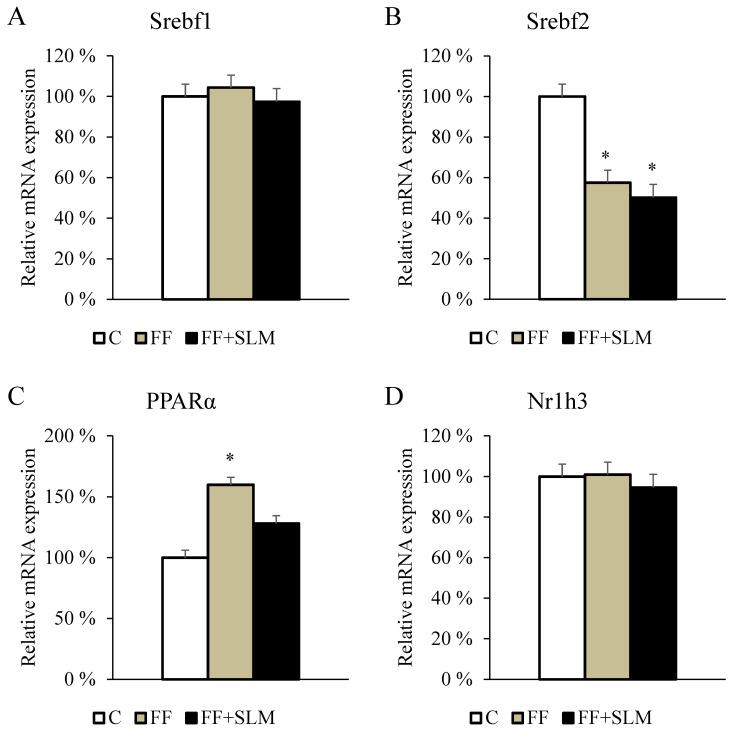
Relative mRNA expression of selected genes involved in lipid metabolism in the liver of HHTg rats treated with FF and combination of FF + SLM compared to untreated HHTg control rats. Data are expressed as means ± SEM; *n* = 6–7 animals per group. Abbreviations: C—control; FF—fenofibrate; FF + SLM—fenofibrate + silymarin. Statistically significant differences: * *p* < 0.05 denotes significance versus the control group.

**Table 1 biomedicines-13-00212-t001:** Oxidative stress parameters in the liver of HHTg rats treated with fenofibrate alone and FF in combination with silymarin.

Parameter	Control	FF	FF + SLM
**Antioxidant system**			
Nrf2 expression (^ΔΔ^Ct)	1.04 ± 0.32	0.68 ± 0.07 *	0.80 ± 0.06
PON-1 (μM/min/mg prot)	6.07 ± 0.67	4.34 ± 0.74 **	6.48 ± 0.98 ##
SOD (U/mgprot)	0.124 ± 0.015	0.107 ± 0.008 *	0.132 ± 0.017 ##
CAT (mM H_2_O_2_/min/mg prot)	1341 ± 213	1848 ± 359 *	1951 ± 431 *
GSH-Px (μM NADPH/min/mgprot)	307 ± 32	301 ± 48	330 ± 62
GR (μM NADPH/min/mg prot)	122 ± 19	136 ± 35	132 ± 22
GST (nM CDNB/min/mg prot)	154 ± 25	194 ± 33 *	226 ± 31 *
GSH (μM/mg prot)	42.47 ± 4.97	39.97 ± 1.40	43.84 ± 11.47
GSSG (μM/mg prot)	3.89 ± 0.46	3.51 ± 0.38	2.93 ± 1.04 *#
GSH/GSSG ratio	11.20 ± 2.56	11.73 ± 1.35	15.37 ± 1.06 *#
**Lipoperoxidation products**			
Conjugated dienes (nM/mgprot)	31.5 ± 5.8	38.0 ± 6.7	38.4 ± 5.9 *
TBARS (nM/mg prot)	1.69 ± 0.44	2.14 ± 0.25 *	1.37 ± 0.32 ##

Data are expressed as means ± SEM; *n* = 6–7 animals per group. Abbreviations: FF—fenofibrate; FF + SLM—fenofibrate + silymarin; Nrf2—nuclear factor erythroid 2-related factor 2; PON-1—paraoxonase-1; SOD—superoxide dismutase; CAT—catalase; GSH-Px—glutathione peroxidase; GR—glutathione reductase; GST—glutathione transferase; GSH—reduced form of glutathione; GSSG—oxidized form of glutathione; TBARS—thiobarbituric reactive substances. Statistically significant differences: * *p* < 0.05; ** *p* < 0.01 denotes significance versus control; # *p* < 0.05; ## *p* < 0.01 denotes significance versus FF.

**Table 2 biomedicines-13-00212-t002:** Oxidative stress parameters in the heart of HHTg rats treated with fenofibrate alone and with combination with silymarin.

Parameter	Control	FF	FF + SLM
**Antioxidant system**			
SOD (U/mg prot)	0.071 ± 0.011	0.069 ± 0.009	0.065 ± 0.008
GSH-Px (μM NADPH/min/mg prot)	128 ± 11	120 ± 13	162 ± 10 *#
GR (μM NADPH/min/mg prot)	42 ± 4	55 ± 4 *	55 ± 1.8 *
GST (nM CDNB/min/mg prot)	31 ± 3	36 ± 5	46 ± 4 *
CAT (mM H_2_O_2_/min/mg prot)	486 ± 38	521 ± 39	510 ± 35
GSH (μM/mg prot)	17 ± 1.3	18.4 ± 0.9	18.6 ± 1.1
**Lipoperoxidation products**			
Conjugated dienes (nM/mg prot)	31.5 ± 5.8	38.0 ± 6.7 *	38.4 ± 5.9 *
TBARS (nM/mg prot)	0.57 ± 0.03	0.59 ± 0.03 *	0.47 ± 0.05 #

Data are expressed as means ± SEM; *n* = 6–7 animals per group. Abbreviations identical to those in Table 2. Statistically significant differences: * *p* < 0.05 denotes significance versus control; # *p* < 0.05 denotes significance versus FF.

**Table 3 biomedicines-13-00212-t003:** Oxidative stress parameters in the kidney cortex of HHTg rats treated with fenofibrate alone and with combination with silymarin.

Parameter	Control	FF	FF + SLM
**Antioxidant system**			
SOD (U/mgprot)	0.039 ± 0.004	0.043 ± 0.05	0.057 ± 0.06 *
GSH-Px (μM NADPH/min/mg prot)	111 ± 9	95 ± 7	142 ± 12 #
GR (μM NADPH/min/mg prot)	44 ± 9	53 ± 3 *	40 ± 3 #
GST (nM CDNB/min/mg prot)	56 ± 4	78 ± 4 *	77 ± 7 *
CAT (mM H_2_O_2_/min/mg prot)	600 ± 39	576 ± 19	601 ± 62
GSH (μM/mg prot)	22.9 ± 1.4	24.0 ± 1.6	18.8 ± 1.6
**Lipoperoxidation products**			
Conjugated dienes (nM/mg prot)	19.7 ± 1.5	38.0 ± 2.6 *	21.9 ± 2.6 #
TBARS (nM/mg prot)	0.52 ± 0.02	0.66 ± 0.05 *	0.58 ± 0.04

Data are expressed as means ± SEM; *n* = 6–7 animals per group. Abbreviations identical to those in Table 2. Statistically significant differences: * *p* < 0.05 denotes significance versus control; # *p* < 0.05 denotes significance versus FF.

**Table 4 biomedicines-13-00212-t004:** Effects of FF alone and in combination with SLM on serum inflammation parameters.

Parameter	Control	FF	FF + SLM
MCP-1 (ng/mL)	3.36 ± 0.85	1.96 ± 0.31 **	2.38 ± 0.58 #
IL-6 (pg/mL)	125 ± 34	105 ± 46	97 ± 45
CRP (μg/mL)	324 ± 39	287 ± 78	279 ± 40

Data are expressed as means ± SEM; *n* = 6–7 animals per group. Abbreviations: FF—fenofibrate; FF + SLM—fenofibrate + silymarin; protein; MCP-1—monocyte chemoattractant protein-1; IL-6—interleukin 6; CRP—C reactive protein. Statistically significant differences: ** *p* < 0.01 denotes significance versus control; # *p* < 0.05 denotes significance versus FF.

## Data Availability

Data is contained within the article.

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
