# Peer review of "Beneficial Effect of Fenofibrate in Combination with Silymarin on Parameters of Hereditary Hypertriglyceridemia-Induced Disorders in an Animal Model of Metabolic Syndrome"

_biomedicines, 2025, doi:10.3390/biomedicines13010212_

Round 1

Reviewer 1 Report

Comments and Suggestions for Authors

This manuscript studied is Beneficial Effect of Fenofibrate in Combination with Silymarin on Hypertriglyceridemia-Induced Disorders in an Animal Model of Metabolic Syndrome. This work is interesting, which is a significant advancement over existing knowledge, but it needs improvements before considering for publication. The publication is recommended, subjected to revision as mentioned below in comments to the authors:

1.     The abstract should be concise, precise, and accurate. The abstract should also include results in the form of values.

2.     Please provide a Graphical abstract of the main findings of the study.

3.     Please, add the innovative aspects of this study.

4.     How many times were the experiments repeated?

5.     Please replace “H2O2” with “H2O2” In the text.

6.     Please add the chemical formulas for the different reagents used namely ammonium molybdate and potassium dihydrogen phosphate

7.     Please add the year of the Statistica software.

8.      “ml” should replace with “mL”, please check whole manuscript and revise it. Please check all units in the manuscript and revise them in the same style.

9.     Page 4, line 182 please add the H2SO4 concentration.

10.  In the section 2.7. Glucose utilization in adipose tissue: The pieces of tissues were homogenized, and lipids were extracted (chloroform:methanol 2:1) at 4 °C overnight, then KH2PO4 was added, an aliquot was  evaporated, reconstituted in scintillation liquid, and the radioactivity measured by scintillation counting.

Why did you perform the extraction with the chloroform:methanol mixture with a ratio of 2:1? What is the exact value of extraction time? The concentration of KH2PO4? What is the role of the latter?

11.  In Figure 6, please add the axis titles.

Reviewer 2 Report

Comments and Suggestions for Authors

The topic was found to be relevant to the field and the manuscript is in the SCOPE of the journal. It is a continued study.  The authors had combined two molecules and tried to study their effect.  So it makes it a formulation kind of study or merely to check the effect of a combination.

Studies require more pieces of evidence or more comparative studies from previous research to justify the claims.

In the abstract data can be added like how many fold decrease or increase. 

Fig 1 F H I J statistical analysis missing

Fig 3 ** indicated but missing in figure kindly remove it

Need more statistics to understand the data.  few places statistics are missing i.e CAT, GST, GR studies.

GSH, GR statistics missing and same in other tables. 

The author needs to check the statistics carefully. 

Comments on the Quality of English Language

Nil

Reviewer 3 Report

Comments and Suggestions for Authors

In abstract, mention the research finding and implications in the last part of the study.

The introduction covers hypertriglyceridemia and its risks in detail. The concept of fenofibrate (FF) and silymarin (SLM) combined therapy should be strongly linked to the research gap. Rewrite the paragraph starting at line 88 to clarify this study's uniqueness and explanation.

The explanation of the animal model (HHTg rats) is comprehensive; however, the methodology for incorporating FF and SLM need further explanation, especially with dosage selection. Include a concise reason for the utilization of 1% micronized silymarin (Line 123).

Figure 1 fails to elucidate the potential physiological significance of the observed changes in adipose tissue mass.

Discussion regarding FF's failure to diminish ectopic triglyceride accumulation (Line 444) needs to encompass wider comparisons with alternative PPARα agonists. This can offer more context for the results.

The therapeutic efficacy of SLM is impaired by its low absorption and rapid excretion, potentially diminishing its effectiveness as a complementary therapy (Page 3).

The investigation was conducted on a small number of animals, which may restrict the generalizability of the results. Need to properly explain.

The conclusion is detailed but should underscore future directions, particularly the prospective clinical use of FF+SLM therapy in human models.Add more details.

Round 2

Reviewer 2 Report

Comments and Suggestions for Authors

Nil

Reviewer 3 Report

Comments and Suggestions for Authors

No further issue.